schizophrenia; community-based rehabilitation; psychosocial interventions; community mental health services; malnutrition

**Corresponding author:**
Laura Asher;
Email: Laura.asher@nottingham.ac.uk

# Community-based rehabilitation intervention for people with schizophrenia in Ethiopia (RISE) cluster-randomised controlled trial: An exploratory analysis of impact on food insecurity, underweight, alcohol use disorder and depressive symptoms

Laura Asher[1] , Rahel Birhane[2,5], Helen A. Weiss[3], Girmay Medhin[4], Medhin Selamu[2,5] , Vikram Patel[6,7] , Mary De Silva[8], Charlotte Hanlon[2,5,9] and Abebaw Fekadu[2,5,10]

[1]Academic Unit of Lifespan and Population Health, School of Medicine, University of Nottingham, Nottingham, UK; [2]WHO Collaborating Centre for Mental Health Research and Capacity-Building, Department of Psychiatry, School of Medicine, College of Health Sciences, Addis Ababa University, Addis Ababa, Ethiopia; [3]MRC International Statistics and Epidemiology Group, London School of Hygiene and Tropical Medicine, London, UK; [4]Aklilu Lemma Institute of Pathobiology, Addis Ababa University, Addis Ababa, Ethiopia; [5]Centre for Innovative Drug Development and Therapeutic Trials for Africa (CDT-Africa), Addis Ababa University, Addis Ababa, Ethiopia; [6]Department of Global Health and Social Medicine, Harvard Medical School, Boston, MA, USA; [7]Harvard T.H. Chan School of Public Health, Boston, MA, USA; [8]The Wellcome Trust, London, UK; [9]Centre for Global Mental Health, Department of Health Service and Population Research, Institute of Psychiatry, Psychology and Neuroscience, King's College London, London, UK and [10]Department of Global Health & Infection, Brighton and Sussex Medical School, Brighton, UK

## Abstract

We evaluated the effectiveness of community-based rehabilitation (CBR) in reducing depressive symptoms, alcohol use disorder, food insecurity and underweight in people with schizophrenia. This cluster-randomised controlled trial was conducted in a rural district of Ethiopia. Fifty-four sub-districts were allocated in a 1:1 ratio to the facility-based care [FBC] plus CBR arm and the FBC alone arm. Lay workers delivered CBR over 12 months. We assessed food insecurity (self-reported hunger), underweight (BMI< 18.5 kg/m2), depressive symptoms (PHQ-9) and alcohol use disorder (AUDIT ≥ 8) at 6 and 12 months. Seventy-nine participants with schizophrenia in 24 sub-districts were assigned to CBR plus FBC and 87 participants in 24 sub-districts were assigned to FBC only. There was no evidence of an intervention effect on food insecurity (aOR 0.52, 95% CI 0.16–1.67; $p$ = 0.27), underweight (aOR 0.44, 95% CI 0.17–1.12; $p$ = 0.08), alcohol use disorder (aOR 0.82, 95% CI 0.24–2.74; $p$ = 0.74) or depressive symptoms (adjusted mean difference − 0.06, 95% CI −1.35, 1.22; $p$ = 0.92). Psychosocial interventions in low-resource settings should support access to treatment amongst people with schizophrenia, and further research should explore how impacts on economic, physical and mental health outcomes can be achieved.

## Impact statement

This article presents exploratory findings from the RISE trial, which is to the best of our knowledge the first randomised controlled trial of any psychosocial intervention for people with schizophrenia in a low-income country setting. On the basis of the main RISE trial results, we recommended psychosocial interventions such as community-based rehabilitation (CBR) should be delivered as an adjunct to facility-based care (FBC), and avenues for the implementation of this (e.g., to consider a new cadre of mental health worker) were proposed. We also proposed a minimum 12-month intervention is required. The findings presented in the current analysis do not change those fundamental recommendations, however, we should consider whether the intervention could be modified in their light. CBR is known to increase access to FBC (including medication adherence and attendance to clinic appointments), and previous research in Ethiopia suggests this care can improve depressive symptoms, alcohol use disorder and food insecurity. We suggest that facilitating access to care, for example through adherence support, help to access free medication and appointment reminders, should be developed as a key focus of psychosocial interventions to ensure all people with schizophrenia experience these benefits. However, these approaches should be delivered in parallel to efforts to support social inclusion and livelihoods. Future research could investigate the acceptability and impact of

feasible models of livelihood support, for example, using savings groups, rather than microfinance schemes requiring external capital. The findings of this study may be useful to researchers and practitioners in developing new models of community-based psychosocial support for people with severe mental health conditions in low and middle-income countries.

## Introduction

The potentially serious impacts of schizophrenia on functioning, livelihoods, and mortality are well-documented globally (Asher et al., 2017a, 2018a; Tirfessa et al., 2019). Alcohol use disorder, depression, undernutrition and household food insecurity are common and interlinked states amongst people with schizophrenia in low-income countries that may be modified by access to psychosocial support. Co-morbid alcohol use disorder was estimated to affect 29% of people with schizophrenia in a rural district in Ethiopia (Hanlon et al., 2019). As well as increasing the risk of all-cause mortality (Correll et al., 2022), alcohol use disorder is associated with higher rates of suicide, incarceration, homelessness, physical health problems, and treatment drop out amongst people with schizophrenia (Bouchard et al., 2022). Over 50% of people with schizophrenia also experience depression and this is associated with poor quality of life, longer illness duration and more frequent psychotic episodes, substance use, as well as suicide (Gregory et al., 2017). Amongst people with severe mental illness in Ethiopia, depression and suicide attempts are more common in those with poorer functioning, poorer quality of life and alcohol or substance use disorder (Shibre et al., 2014; Fanta et al., 2020). In rural Ethiopia, severe household food insecurity, an important marker of poverty, was experienced by 33% of people with schizophrenia in a rural district (double that found in the general population (Tirfessa et al., 2017). This effect is likely to be due to difficulties pursuing livelihood activities, such as farming, in the face of functional impairments in people with schizophrenia, and time-consuming caregiving activities amongst family members (Tirfessa et al., 2019). Undernutrition, poorer living conditions, increased susceptibility to infectious diseases, and inability to access healthcare are all potential sequalae of household food insecurity (Berrington de Gonzalez et al., 2010; Teferra et al., 2011; Hailemichael et al., 2019).

In Ethiopia, task-shared integrated care delivered by non-physicians in primary healthcare is safe and effective in reducing disability, symptoms, alcohol use disorder, depressive symptoms, suicide attempts, household food insecurity, as well as physical restraint, and discrimination in people with schizophrenia (Hanlon et al., 2019; Tirfessa et al., 2020; Hanlon et al., 2022). Psychosocial interventions and strategies such as community-based rehabilitation (CBR) are recommended to address the complex social, economic and clinical needs of individuals who have not responded to standard care in low and middle-income country (LMIC) contexts. We designed the Community-based Rehabilitation Intervention for people with Schizophrenia in Ethiopia (RISE) intervention to meet the needs of this group in a rural Ethiopian district (Asher et al., 2015). The primary aim of CBR was to improve functioning, that is to support individuals to develop the skills and confidence to perform their previous or desired roles and activities. We developed a theory of change to delineate the pathways through which we hypothesised improvements in functioning could be achieved (Asher et al., 2018b; Supplementary File 1). We identified intermediate outcomes to be: 'improved understanding about mental illness and human rights', 'improved family stability and care', 'increased access to mental health care', 'reduced stigma and abuse', 'increased social inclusion', 'improved physical health', 'increased

income', 'reduced symptoms', 'increased involvement in decision-making' and 'increased self-esteem and hope' (Asher et al., 2018b). We therefore designed the RISE CBR intervention to address all five pillars of the WHO's CBR matrix: health, social, livelihoods, empowerment and education, to the extent possible with available resources (Asher et al., 2015). A broad range of needs could be addressed through optional modules including, for example, Improving physical health (health domain), Support returning to work (livelihoods domain), Dealing with stigma (empowerment domain), Taking part in community life (social domain) and Improving literacy (education domain), along with an underpinning emphasis on improving self-esteem and fostering a sense of hope. A structured community mobilisation component, including public awareness-raising talks, was delivered in parallel. We hypothesised that causal effects would likely be bi-directional with, for example, improved functioning predicted to positively impact on physical health. This could be due to increased farming activity resulting in greater food availability and improved nutrition; and also, greater income leading to poverty reduction.

The main RISE trial analysis, which reported the pre-specified main outcomes, demonstrated that CBR as an adjunct to facility-based care (FBC) is effective in reducing disability at 12 months in persons with schizophrenia who had disabling illness after 6 months of access to standard care (Asher et al., 2022). There were also beneficial impacts on symptom severity, caregiver tension and worrying, and in increasing anti-psychotic medication adherence and attendance to FBC. The impact of clinical or psychosocial interventions on nutritional status, substance use or depressive symptoms in people with schizophrenia is rarely evaluated in LMIC, with the focus tending to be on functioning and symptoms (Chatterjee et al., 2014; Asher et al., 2017b; Li et al., 2018; Jordans et al., 2019; Shidhaye et al., 2019). In this article, we explore the effectiveness of CBR on the exploratory outcomes of depressive symptoms, alcohol use disorder, underweight and household food insecurity. The aim of the current analysis is to aid in refining CBR for future implementation by indicating its potential to influence these vulnerabilities.

## Methods

### Study design and participants

The RISE cluster-randomised controlled trial was conducted in Sodo district, Gurage Zone, Southern Nations, Nationalities and Peoples' Region in Ethiopia. The study protocol and main trial results were previously published (Asher et al., 2016, 2022). Sodo district has a total population of 170,000 people in 58 sub-districts. There are high levels of poverty and the main economic activity is subsistence farming and small-scale trading. Most of the population live in remote rural areas. Primary care is delivered by nurses and health officers at one primary hospital and seven health centres. Care costs are usually out-of-pocket with a fee waiver available for the poorest. The Ethiopian government does not provide social protection, such as disability grants or food subsidies, nor employment opportunities, for people with mental illness. There are also

no non-governmental organisations providing such support in the study area.

Randomisation units were sub-districts of Sodo district. In total, 54 sub-districts were available after subtracting the four pilot sub-districts (Asher et al., 2018b; Figure 1). Sub-districts were excluded if no eligible participants were identified. People with suspected schizophrenia were identified by the Programme for Improving Mental Healthcare (PRIME) study in Sodo through key community informants who had received half a day of training in the typical presentations of schizophrenia (Hanlon et al., 2019). Trained psychiatric nurses then used the Operational Criteria for Research (OPCRIT) diagnostic interview (McGuffin et al., 1991) (which applies Diagnostic and Statistical Manual of Mental Disorders fourth edition (DSM-IV) criteria), and those with a confirmatory diagnosis of schizophrenia were recruited to the PRIME study. FBC was available for people in the PRIME study with a confirmed

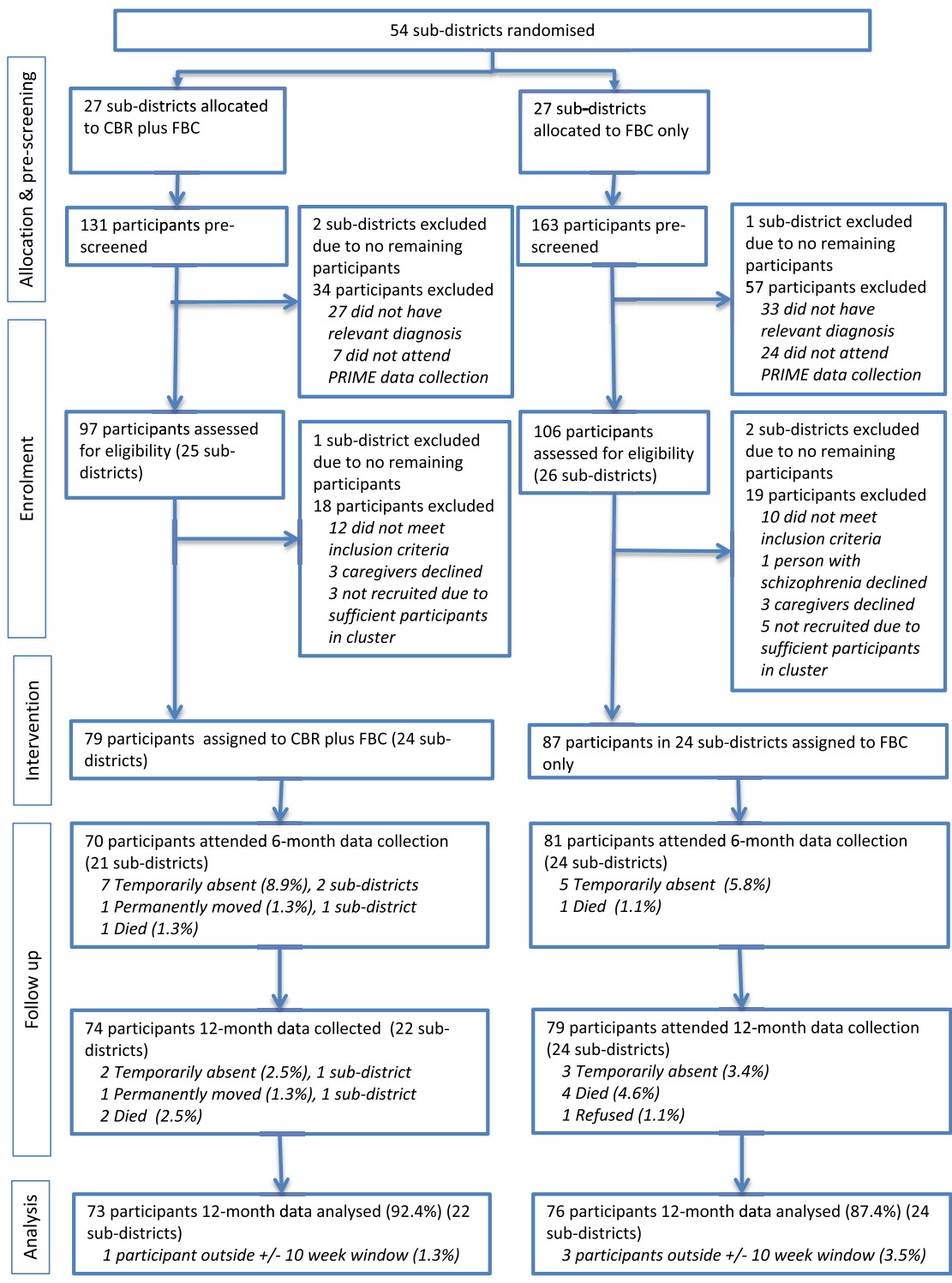

**Figure 1.** Trial profile

diagnosis of schizophrenia. The PRIME study started 6 months before RISE trial recruitment. RISE study participants had therefore had the opportunity to access FBC for 6 months before being enrolled.

RISE participants were recruited from two sources: (i) individuals who were participating in the PRIME study and (ii) individuals who had not participated in the PRIME study but had been identified by PRIME as having suspected schizophrenia. These participants had not attended FBC. Participants from the PRIME study were recruited at the PRIME 6-month data collection (or on a separate occasion if they did not attend) and individuals who were not participants in the PRIME study were recruited using the PRIME database of people with suspected schizophrenia. PRIME cohort study participants were prescreened by checking PRIME baseline data for a schizophrenia diagnosis. Participants meeting this criterion underwent full RISE eligibility assessment by the trial nurse using PRIME cohort 6-month data. Participants not in the PRIME cohort had an initial consent taken, before data was collected on the eligibility instruments, including the OPCRIT interview. The inclusion criteria were: In PRIME cohort study or not engaged in FBC but resident in Sodo district; diagnosis of schizophrenia, schizoaffective disorder or schizophreniform disorder; ≥18 years; resident in sub-district for 6 months and intending to stay; had a primary caregiver willing to participate; and ≥ 1 marker/s of severe, disabling or enduring illness (Brief Psychiatric Rating Scale-Expanded [BPRSE] score ≥ 52; or proxy or self-rated 36-item WHO Disability Assessment Schedule [WHODAS] 2·0 score ≥ 35; or continuous illness last 6 months; or symptomatic ≥3 of last 6 months or Clinical Global Impression [CGI] severity score ≥ 3).

### Randomisation and masking

Randomisation was carried out before participant recruitment by an independent statistician using Stata version 14 (StataCorp, 2015). A total of 54 subdistricts were randomised in a 1:1 ratio to either the intervention (CBR plus FBC group) or to the control group (FBC alone group). To prevent imbalance for potential confounding factors the statistician employed an optimization algorithm calculated for subdistrict mean WHODAS score at PRIME baseline and the potential number of participants per sub-district. They applied the procedure for each health centre and used a computer programme to randomly chose the allocation sequence from the set of optimal ones. The statistician was masked to the intervention/control label. Researchers responsible for identifying and recruiting participants and collecting outcome data were masked to allocation status. All co-authors (except RB) remained masked until the final analysis was complete.

### Procedures

FBC is a stepped care model. The majority of care was delivered in primary care, and comprised prescription of anti-psychotic medication and psychoeducation by nurses and health officers trained for 2 weeks in the WHO mental health Gap Action Programme-Intervention Guide (mhGAP-IG) and supervised by a psychiatric nurse. Primary care staff could refer individuals to psychiatric nurse-led out-patient care (secondary care) or psychiatrist-led in-patient care (tertiary care) as required. Frequency of contact with FBC was determined by clinical need. Health extension workers in each sub-district were trained in awareness-raising on causes and treatment of mental disorders. A district-level community-

advisory board oversaw implementation of the integrated FBC model.

CBR was delivered by 11 CBR workers, each covering a catchment area of one, two or three sub-districts, linked to a health centre. CBR delivery commenced immediately after trial recruitment and lasted 12 months. CBR workers supported a median of seven participants (range 4–11). CBR workers were lay persons recruited from the local area with at least 10 years of education but no prior experience in delivering mental healthcare. They received 5 weeks initial training in CBR delivery, guided by a manual, including basic counselling and problem-solving techniques (Asher et al., 2021). This was followed by monthly half-day top-up training during intervention delivery. Training was split between classroom teaching (role-plays, discussions, videos, and quizzes) and fieldwork (shadowing psychiatric nurses and CBR workers for children with disabilities). In the pre-trial pilot study, CBR workers each delivered CBR to one family for 6 months (Asher et al., 2018b). CBR workers were expected to work full-time for RISE and their salary was matched to that of local health extension workers.

CBR visits lasted 30–90 min and took place at the participants' home. The intervention emphasised human rights, social inclusion and the possibility of recovery. Phase 1 lasted 2–3 months, during which time home visits were every 1–2 weeks. The focus was on engagement and addressing core needs through the modules 'Understanding schizophrenia', 'Improving access to health services', 'Preparing for a crisis' and 'Dealing with human rights problems'. In Phase 2, lasting 5–6 months, home visits were every 2 weeks. A subset from 11 optional modules were selected to support achievement of individual goals. Optional modules included 'Improving physical health', which included simple advice and problem-solving steps related to healthy eating, reducing alcohol consumption and accessing physical healthcare. The 'Dealing with stress and anger' module included relaxation techniques and simple anger management approaches. The 'Getting back to work' module covered preparation and problem solving for caregivers to support their relative, focusing on returning to farm work. Other optional modules covered adherence support, family intervention, support returning to social activities, and dealing with stigma. In Phase 3 (4 months), the emphasis was on preventing relapse and maintaining progress. CBR workers met with community members to mobilise resources for individual participants, for example, treatment costs or family mediation. At a sub-district level, CBR workers conducted additional community mobilisation (public awareness-raising meetings and engaging with community leaders) and ran family support groups. Two supervisors, who were not mental health specialists, oversaw the frequency, content and quality of home visits. If the CBR workers identified suicide intent, relapse, physical illness or medication side-effects they referred participants to the health centre in addition to regular appointments. CBR workers were referred to the trial psychiatric nurse if participants were unable to attend the health centre.

### Outcomes

- Depressive symptoms were measured using a version of the Patient Health Questionnaire (PHQ-9). The PHQ-9 has been demonstrated to have good construct (Cronbach's alpha 0.87), concurrent and convergent validity in the study district (Habtamu et al., 2022).
- Alcohol use disorder was assessed using a version of the Alcohol Use Disorders Identification Test (AUDIT) adapted for the

Ethiopian setting. An AUDIT score ≥ 8 indicates probable alcohol use disorder (hazardous or harmful drinking). The AUDIT has been demonstrated to have good internal consistency in the study district (Cronbach's alpha 0.84) (Zewdu et al., 2019)

- Underweight was measured using body mass index (weight (kg)/ (height, in m)$^2$ < 18.5.
- Food insecurity was assessed using a single binary question on hunger due to lack of resources or food in last month.

Data were collected by lay data collectors at baseline, 6 months (±10 weeks) and 12 months (±10 weeks) at the participant's home or health centre. Sociodemographic information was collected at the PRIME cohort baseline or RISE baseline.

### Statistical analysis

Assuming 23% attrition, we aimed to recruit 182 participants to provide an endline sample size for the analysis of 140 participant dyads in 54 sub-districts. This sample size provides 85% power to detect a 20% absolute difference in the primary outcome, WHODAS score, between treatment arms with 5% significance, assuming a mean WHODAS score of 50 in the control arm, a k (coefficient of variation) of 0·14 and a within-cluster standard deviation (SD) of 16. The study was not powered to detect effects in the exploratory outcomes reported in the current article. Baseline characteristics of participants and sub-districts were compared between treatment arms. Outcome analyses used intention-to-treat principles, analysing participants according to the arm to which they were randomised. Outcome measures were summarised at baseline, 6-month and 12-month data points by treatment arm. We conducted a repeated measures analysis combining 6 and 12 month outcome data. We also analysed the 6 and 12 month data separately as a sensitivity analysis. For binary outcomes, we reported intervention effects as minimally- and fully-adjusted odds ratios (aORs) and 95% CIs estimated from logistic random effect regression models. For continuous outcomes, we estimated intervention effects using linear mixed-effects regression and reported them as minimally-adjusted mean differences, fully-adjusted mean differences (FAMDs), and effect sizes defined as standardised mean differences, with 95% CIs. Minimally adjusted models included baseline proxy-rated WHODAS total score, health centre (fixed effect) and sub-district (random effect). CBR worker was not adjusted for due to the high collinearity of CBR worker and health centre. Fully adjusted models included variables associated with missingness (threshold $p < 0.1$; presented previously; Asher et al., 2022), and variables deemed unbalanced between arms at baseline. The analyses were complete case (i.e., we only included participants with data on the variables of interest). Process and adverse event data have been presented previously (Asher et al., 2022). Statistical analyses were done with Stata version 15. An independent Data Safety and Monitoring Board oversaw the study. The trial is registered with ClinicalTrials.gov (NCT02160249).

### Results

Participants were enrolled between Sept 16, 2015 and Mar 11, 2016 (Figure 1). Of the 54 available sub-districts, 27 were randomised to the intervention arm and 27 to the control arm. A total of 294 potential participants were pre-screened, of whom 91 were excluded. A further 37 individuals were not enrolled. Of these, 22 participants (10·8%) did not meet the inclusion criteria, one participant and six caregivers declined (3·4%) and eight participants were excluded due to sufficient numbers in the cluster (3·9%). Three sub-districts were

excluded at each of the pre-screening and enrolment stages. Hence of 54 potential sub-districts for inclusion, 48 were included. Twenty-four sub-districts (79 participants) were assigned to the intervention arm and 24 sub-districts (87 participants) were assigned to the control arm.

Participants had a mean age of 31·4 years (range 18–80). At baseline, there were high levels of disability (mean proxy-rated WHODAS 51·5 [SD 23·6]) and unemployment (57.8%). Alcohol use disorder was detected in 20.0% of participants. Underweight and food insecurity were found in 34.6% and 15.1% of participants. We observed some imbalance in baseline participant characteristics by arm (Table 1). Control-arm participants were more likely than intervention arm participants to be female, to have lower household socio-economic status, to be unemployed, and to have social

**Table 1.** Baseline characteristics of RISE trial participants by treatment group

| Persons with schizophrenia | Facility-based care group (*n* = 87) | CBR plus facility-based care group (*n* = 79) |
|---|---|---|
| Sex (*n* (%)) | | |
| Male | 51 (58.6%) | 52 (65.8%) |
| Female | 36 (41.4%) | 27 (34.2%) |
| Age (years) (median (IQR)) | 33 (25, 40) | 30 (25, 45) |
| Marital status (*n* (%))[a] | | |
| Single | 41/80 (51.3%) | 38/75 (50.7%) |
| Has a partner (married, married not living together) | 26/80 (32.5%) | 25/75 (33.3%) |
| Separated/Divorced/ Widowed | 13/80 (16.3%) | 12/75 (16%) |
| Occupation (*n* (%)) | | |
| No occupation | 15 (17.2%) | 9 (11.4%) |
| Home worker | 33 (37.9%) | 27 (34.2%) |
| Unskilled labourer | 34 (39.1%) | 42 (53.2%) |
| Other | 5 (5.8%) | 1 (1.3%) |
| Education status (*n* (%))[a] | | |
| No formal education | 46/80 (57.5%) | 36/75 (48.0%) |
| Primary education | 22/80 (27.5%) | 35/75 (46.7%) |
| Secondary education and above | 12/80 (15.0%) | 4/75 (5.3%) |
| Socio economic status (*n* (%))[a] | | |
| Higher (poverty index ≤3) | 42/80 (52.5%) | 47/74 (63.5%) |
| Lower (poverty index >3) | 38/80 (47.5%) | 27/74 (36.5%) |
| Residence (*n* (%))[a] | | |
| Urban | 11/80 (13.8%) | 8/74 (10.8%) |
| Rural | 69/80 (86.3%) | 66/74 (89.2%) |
| Travel time to nearest health facility (*n* (%))[a] | | |
| ≤ 60 min | 51/80 (63.8%) | 48/75 (64.0%) |
| 61 to 120 min | 13/80 (16.3%) | 17/75 (22.7%) |
| ≥121 min | 16/80 (20.0%) | 10/75 (13.3%) |
| Diagnosis (*n* (%)) | | |
| Schizophrenia | 70 (80.5%) | 68 (86.1%) |

*(Continued)*

**Table 1.** (Continued)

| Persons with schizophrenia | Facility-based care group (n = 87) | CBR plus facility-based care group (n = 79) |
|---|---|---|
| Schizoaffective/ schizophreniform disorder | 17 (19.5%) | 11 (13.9%) |
| Duration illness in years (median (IQR))[a] | 4 (1.9, 9) n = 64 | 3.7 (1.5, 7.3) n = 55 |
| Co-morbid medical disorder (n (%))[a] | | |
| No | 69/76 (90.8%) | 69/72 (95.8%) |
| Yes | 7/76 (9.2%) | 3/72 (4.2%) |
| Proxy-rated total WHODAS (mean (SD)) | 52.6 (23.6) | 50.2 (23.6) |
| BPRS-E total (mean (SD)) | 47.2 (13.4) n = 85 | 48.9 (14.2) n = 75 |
| CGI (n (%)) | | |
| Normal or borderline | 6 (6.9%) | 2 (2.5%) |
| At least mildly ill (score ≥ 3) | 81 (93.1%) | 77 (97.5%) |
| Illness course last 6 months (LCS) (n (%)) | | |
| Episodic | 3 (3.5%) | 4 (5.1%) |
| Continuous | 66 (75.9%) | 66 (83.5%) |
| Never psychotic | 18 (20.7%) | 9 (11.4%) |
| Antipsychotic medication adherence (n (%)) | | |
| All or most of time | 35/83 (42.2%) | 36/77 (46.8%) |
| Sometimes, occasionally or not at all | 48/83 (57.8%) | 41/77 (53.3%) |
| Engagement with care (n (%)) | | |
| No healthcare attendance and no medication adherence | 32/83 (38.6%) | 30/77 (39.0%) |
| Either healthcare attendance or medication adherence | 23/83 (27.7%) | 17/77 (22.1%) |
| Healthcare attendance and medication adherence | 28/83 (33.7%) | 30/77 (39.0%) |
| Alcohol use disorder (AUDIT total ≥ 8) (n (%)) | | |
| No | 70/86 (81.4%) | 58/74 (78.4%) |
| Yes | 16/86 (18.6%) | 16/74 (21.6%) |
| Depression (PHQ-9 mean (SD)) | 10.1 (5.3) | 10.0 (5.1) |
| Food insecurity (n (%)) | | |
| No | 74 (85.1%) | 67 (84.8%) |
| Yes | 13 (14.9%) | 12 (15.2%) |
| Underweight (BMI < 18.5) (n (%)) | | |
| No | 47 (68.1%) | 38 (62.3%) |
| Yes | 22 (31.9%) | 23 (37.7%) |
| Restrained last 6 months (n (%)) | | |
| No | 82 (94.3%) | 75 (94.9%) |
| Yes | 5 (5.8%) | 4 (5.1%) |
| Any experience of discrimination last 6 months (n (%)) | | |
| No | 40 (46.0%) | 38 (48.1%) |
| Yes | 47 (54.0%) | 41 (51.9%) |

(Continued)

**Table 1.** (Continued)

| Persons with schizophrenia | Facility-based care group (n = 87) | CBR plus facility-based care group (n = 79) |
|---|---|---|
| Unemployed (n (%)) | | |
| No | 32 (36.8%) | 38 (48.1%) |
| Yes | 55 (63.2%) | 41 (51.9%) |
| Social support (n (%)) | | |
| Poor | 23 (26.4%) | 35 (44.3%) |
| Intermediate | 49 (56.3%) | 31 (39.2%) |
| Strong | 15 (17.2%) | 13 (16.5%) |
| **Caregiver** | | |
| Mean total IEQ score (mean (SD)) | 40.7 (19.4) | 40.1 (16.0) |
| Depression (PHQ-9 ≥ 5) (n (%)) | | |
| No | 57 (65.5%) | 36 (45.6%) |
| Yes | 30 (34.5%) | 43 (54.4%) |
| Unemployed (n (%)) | | |
| No | 55 (63.2%) | 43 (54.4%) |
| Yes | 32 (36.8%) | 36 (45.6%) |

| Sub-district | Facility-based care group (n = 24) | CBR plus facility-based care group (n = 24) |
|---|---|---|
| Location (n (%)) | | |
| Urban | 1 (4.2%) | 2 (8.3%) |
| Rural | 23 (95.8%) | 22 (91.7%) |
| Baseline number of participants (median (IQR)) | 3 (1.5, 4.5) | 2.5 (1, 5) |
| Proxy-rated total WHODAS (median (IQR)) | 52.8 (45.2, 61.1) | 54.1 (41.4, 62.4) |

[a]Data collected at PRIME baseline.

support. Control arm caregivers were less likely to be unemployed or be depressed than intervention arm caregivers. At 12 months (±10 week protocol-defined window), depressive symptoms, alcohol use and food insecurity outcome data were available for 73 (92·4%) participants (22 clusters) in the intervention arm and 76 (87·4%) participants (24 clusters) in the control arm (Figure 1). 12-month BMI outcome data were available for 64 (81.0%) participants (19 clusters) in the intervention arm and 58 (66.7%) participants (22 clusters) in the control arm.

There was no evidence of an intervention effect on food insecurity (aOR 0.52, 95%CI 0.16 to 1.67; *p* = 0.27), underweight (aOR 0.44, 95%CI 0.17 to 1.12; *p* = 0.08), alcohol use disorder (aOR 0.82, 95%CI 0.24 to 2.74; *p* = 0.74), or depressive symptoms (adjusted mean difference − 0.06, 95%CI -1.35, 1.22; *p* = 0.92) combining 6 and 12 month data (Table 2), nor any effect at 6 or 12 months separately (Supplementary File 2).

## Discussion

This article presents exploratory findings from the RISE trial, which is to the best of our knowledge the first randomised

controlled trial of any psychosocial intervention for people with schizophrenia in a low-income country setting. CBR was a complex intervention with a range of components including efforts to improve livelihoods and support to improve physical health. Depressive symptoms, alcohol use disorder, underweight and food insecurity, whilst they were not prioritised as main outcomes, were therefore worthy of the exploratory analysis presented in this article. Moreover, excess mortality is a key argument for the prioritisation of schizophrenia in global mental health efforts (Patel, 2015). In Ethiopia, reflecting global patterns (Ali et al., 2022), the risk of premature mortality amongst people with schizophrenia is more than double that of the general population over a 10 year follow-up period (Fekadu et al., 2015), with infectious diseases, suicide, accidents and severe malnutrition the most common causes of death (Teferra et al., 2011; Fekadu et al., 2015). Alcohol use disorder, depression, undernutrition and household food insecurity may arguably represent modifiable risk factors for premature mortality. We did not find any impact of CBR on any of these outcomes. These findings build on the main trial results which showed an impact on disability, symptoms and access to mental health care, along with high levels of intervention fidelity; and which in turn reflected the benefits of community-based psychosocial interventions found in middle-income country settings (Chatterjee et al., 2014; Asher et al., 2017b; Li et al., 2018; Luo et al., 2019; Chen et al., 2020). As such, whilst the intended improvements in functioning materialised, these did not appear to be achieved through the hypothesised pathways of improved physical health and increased income delineated on the theory of change (Supplementary File 1). Future qualitative analyses, drawing on data from CBR participants and workers, will further interrogate the theory of change, including exploration of barriers and facilitators to the success of CBR and its mechanisms of impact.

The prevalence of underweight in male RISE participants was greater than the general population in Ethiopia, confirming that malnutrition may be an area for need for this group (women: 22% in general population age 18–49 years vs. 22% in RISE; men: 33% general population age 18–49 years vs. 43% in RISE (ICF, 2016)). The PRIME cohort study, in which RISE was nested, demonstrated that in rural Ethiopia access to mental health care appears to reduce food insecurity amongst people with schizophrenia, through effects on work functioning (Tirfessa et al., 2020). We have previously reported the lack of benefits of CBR on work functioning and employment when delivered in addition to FBC (Asher et al., 2022), and this may explain the absence of effect on underweight and food insecurity found in the current analysis. It is likely that there are insufficient opportunities in this rural Ethiopian context to make meaningful changes to employment. The absence of a savings or microfinance component may also have limited the potential impact on food insecurity. Furthermore, the RISE pilot study demonstrated that due to high poverty levels in the wider community, financial or material support for participants, mobilised by CBR workers, was usually precarious (Asher et al., 2018b). The moral and social support mobilised, for example in the form of family mediation, was typically more available and sustainable. There is also little evidence that the types of simple dietary advice and general physical health advice included in the RISE intervention are effective in improving outcomes for people with schizophrenia in any context (Tosh et al., 2014; Pearsall et al., 2016).

There are indications that access to facility-based mental health care can lead to a reduction in the prevalence of alcohol use disorder amongst people with schizophrenia in Ethiopia. Depressive symptoms also appear to reduce over time with access to care, though these may remit spontaneously (Hanlon et al., 2019). Again, our results show that CBR did not make any additional impacts on these outcomes. The RISE pilot study identified CBR's power to foster a hopeful outlook and improve self-esteem as a common experience of CBR participants, and a potentially important pathway to improved functioning (Asher et al., 2018b). However, our measure of depressive symptoms, PHQ-9, is not designed to capture positive mental health states. Moreover, CBR did not incorporate evidence-based psychological interventions for depression and alcohol use disorder, for example, problem-solving therapy or motivational interviewing. Better understanding of the nature of depression amongst people with schizophrenia in rural Ethiopia, for example, whether it is best understood as an expression of negative symptoms, related to anti-psychotic medication, or distress at social circumstances, would aid the interpretation of our results. However, we did not collect data with the specific aim of interrogating these issues.

An important limitation is that the study was not powered to detect impacts on these exploratory outcomes. Problems with data sparsity and missing BMI data also meant low numbers were included in regression models, making it more difficult to detect any effects. There were large amounts of missing data at 6 months. Further, there was no assessment of the long-term impact of CBR after intervention delivery was completed. Finally, a validated instrument such as the Household Food Insecurity Access Scale (Tirfessa et al., 2020) would have been a preferable measure of food insecurity, compared to the single item assessing hunger.

On the basis of the main RISE trial results, we recommended psychosocial interventions such as CBR should be delivered as an adjunct to FBC, and avenues for the implementation of this (e.g., to consider a new cadre of mental health worker) were proposed (Asher et al., 2022). As we demonstrated an impact on functioning and other outcomes at 12 months, but not 6 months, we also proposed a minimum 12-month intervention is required. The findings presented in the current analysis do not change those fundamental recommendations, however, we should consider whether the intervention could be modified in their light. CBR is known to increase access to FBC (including medication adherence and attendance to clinic appointments), and this care in turn appears to improve depressive symptoms, alcohol use disorder and food insecurity (Hanlon et al., 2019). We suggest that facilitating access to care, for example through adherence support, help to access free medication and appointment reminders, should be developed as a key focus of psychosocial interventions to ensure all people with schizophrenia experience these benefits. However, these approaches should be delivered in parallel to efforts to support social inclusion and livelihoods.

In many middle and high-income countries people living with severe mental illness are eligible for social protection payments, such as disability grants in South Africa (Wright, 2015). There is some evidence that livelihood interventions such as grants delivered through self-help groups might provide additional impacts on poverty when delivered alongside mental healthcare amongst people with schizophrenia (Lund et al., 2013). Such interventions are under-researched but may be challenging to deliver in an affordable, sustainable and scalable way (de Menil et al., 2015). RISE did not include a formal livelihoods intervention such as microfinance due to, (i) worries raised in the intervention development phase that this approach could increase stress and result in the exploitation of participants (Asher et al., 2015), (ii) concerns

**Table 2.** RISE Exploratory outcomes repeated measures analysis

| Outcome | 6 months | | 12 months | | Minimally adjusted analysis | | Fully adjusted analysis | | |
| | Facility-based care group n = 60 | CBR plus facility-based care group n = 52 | Facility-based care group n = 76 | CBR plus facility-based care group n = 73 | Mean difference or odds ratio (95% CI)[a] | p-value | Mean difference or odds ratio (95% CI)[b] | p-value | Effect size (95% CI) |
| --- | --- | --- | --- | --- | --- | --- | --- | --- | --- |
| Food insecurity | 8 (13.3%) | 4 (7.7%) | 6 (7.9%) | 6 (8.2%) | 0.87 (0.35, 2.20)[c] | 0.78 | 0.52 (0.16, 1.67)[c] | 0.27 | – |
| Underweight (BMI < 18.5)[d] | 17 (38.6%) | 12 (31.6%) | 24 (37.5%) | 18 (31.0%) | 0.62 (0.25, 1.51)[e] | 0.29 | 0.44 (0.17, 1.12)[e] | 0.08 | – |
| Alcohol use disorder (AUDIT ≥ 8)[f] | 10 (17.0%) | 9 (17.3%) | 13 (17.1%) | 6 (8.2%) | 0.55 (0.21, 1.51)[g] | 0.25 | 0.82 (0.24, 2.74)[g] | 0.74 | – |
| Depressive symptoms (PHQ-9 score) | 7.1 (4.8) | 7.4 (4.0) | 6.8 (4.4) | 6.3 (4.6) | –0.58 (–1.25, 1.14) | 0.93 | –0.06 (–1.35, 1.22) | 0.92 | 0.01 (–0.23, 0.26) |

*Note:* Data are n (%) or mean (SD) unless otherwise indicated. Continuous outcomes display adjusted mean differences and binary outcomes display odds ratio.
[a]Adjusted for sub-district (cluster) as random effect and health centre and baseline score of outcome as fixed effects.
[b]Adjusted for sub-district (cluster) as a random effect and health centre, baseline score of outcome, baseline disability (proxy-reported total WHODAS), sex, age, residence, baseline socio-economic status, baseline illness course, baseline caregiver burden (IEQ), illness duration, baseline employment status, baseline caregiver employment status, baseline social support, baseline travel time to facility and baseline caregiver depression (PHQ-9) as fixed effects. Illness course and social support reduced to two categories to avoid problems with data sparsity.
[c]Fully adjusted model excludes baseline travel time to facility, residence and socio-economic status due to data sparsity. n = 232 in minimally adjusted and fully adjusted models due to data sparsity in health centre categories.
[d]n = 82 at 6 months and n = 122 at 12 months due to missing data.
[e]Fully adjusted model excludes socio-economic status due to data sparsity. n = 195 in minimally and fully adjusted models due to missing baseline data.
[f]n = 111 at 6 months due to missing data.
[g]Fully adjusted model excludes residence due to data sparsity. n = 241 in minimally adjusted and fully adjusted models due to data sparsity in health centre categories.

that this would not meet the primary need to create an intervention that was scalable using limited resources, and (iii) limited project funds. The feasibility and effectiveness of such initiatives on outcomes for people with schizophrenia is a research and policy priority (Rose-Clarke et al., 2020). Future research could investigate the acceptability and impact of scalable models of livelihood support, for example, using savings groups, rather than microfinance schemes requiring external capital.

Globally there is also a dearth of evidence and guidelines for the treatment of dual diagnosis of schizophrenia and alcohol use disorder (Alsuhaibani et al., 2021). Whilst proposed psychosocial interventions for dual diagnosis include assertive community treatment, cognitive behavioural therapy and motivational interviewing, there is little evidence to support one over another (Hunt et al., 2019). Whilst there is also little evidence for how alcohol use disorder interventions, in general, can be successfully implemented in sub-Saharan Africa (Mushi et al., 2022), a pilot study of brief (one-session) intervention for alcohol use disorder delivered in primary care was found to be acceptable and feasible in community members in rural Ethiopia (Zewdu et al., 2022). Future research could explore the feasibility and impact of similar very brief interventions delivered to people with schizophrenia, either by CBR workers as an optional component, or through referral to primary care. Proposed community-level health promotion initiatives targeting harmful alcohol consumption (Zewdu et al., 2022) could also dovetail with the community mobilisation aspects of CBR.

The most appropriate means to address the high levels of depressive symptoms amongst people with schizophrenia are not clear. Globally, the evidence for anti-depressants for people with schizophrenia remains equivocal (Gregory et al., 2017). Whilst brief and/or trans-diagnostic psychological interventions are effective in improving depression in the general population in LMIC when delivered by lay workers (Weobong et al., 2017; Hamdani et al., 2020), such approaches have not been applied to severe mental illness. Furthermore, lay workers already delivering a time-intensive, complex intervention such as CBR may not have the time capacity to deliver a psychological intervention. Simple problem-solving approaches were also the most challenging skills for CBR workers to acquire (Asher et al., 2021). Screening for suicidality may be a useful way to identify severe depression amongst people with schizophrenia in community settings, followed by referral to a primary health centre for anti-depressants and problem-solving therapy.

## Conclusion

We evaluated the impacts of CBR on a range of exploratory outcomes linked to physical and mental health in people with schizophrenia by conducting a high-quality cluster randomised controlled trial. CBR has previously been shown to improve functioning, symptom severity, and engagement with mental health-care, when provided as an adjunct to FBC. There was no evidence that CBR delivers additional impacts on depressive symptoms, alcohol use disorder, underweight or food insecurity in people with schizophrenia, over and above the beneficial impacts of accessing FBC. Psychosocial interventions in low-resource settings should strengthen efforts to support access to treatment amongst people with schizophrenia alongside social and livelihoods interventions. Future research should explore how further impacts on depressive symptoms, alcohol use, malnutrition and poverty can be achieved.

## List of abbreviations

| | |
|---|---|
| aOR | adjusted odds ratio |
| AUDIT | alcohol use disorder identification test |
| BPRS-E | brief psychiatric rating scale-expanded version |
| CBR | community-based rehabilitation |
| CGI | clinical global impression |
| DSM-IV | Diagnostic and Statistical Manual of Mental Disorders fourth edition |
| LMIC | low and middle-income countries |
| mhGAP-IG | Mental Health Gap Action Programme – Intervention Guide |
| OPCRIT | operational criteria for research |
| PHQ-9 | Patient Health Questionnaire-9 |
| PRIME | Programme for Improving Mental Healthcare |
| RISE | community-based rehabilitation intervention for people with schizophrenia in Ethiopia |
| SD | standard deviation |
| WHODAS | World Health Organisation Disability Assessment Schedule |

**Open peer review.** To view the open peer review materials for this article, please visit http://doi.org/10.1017/gmh.2023.67.

**Supplementary material.** The supplementary material for this article can be found at https://doi.org/10.1017/gmh.2023.67.

**Data availability statement.** The datasets analysed during the current study are available from the corresponding author upon reasonable request.

**Acknowledgements.** We thank the CBR workers and CBR supervisors who supported the implementation of the trial, and the participating individuals.

**Author contribution.** L.A. drafted the report, which all authors reviewed and approved. L.A., A.F., C.H., M.D.S., G.M., H.A.W. and V.P. designed the trial. L.A., A.F., C.H., M.D.S., R.B. and V.P. devised the intervention content and data collection instruments. L.A., A.F. and R.B. were responsible for trial conduct. L.A. and R.B. were responsible for database design and management. L.A. and H.A.W. did the statistical analyses. L.A. and H.A.W. have verified the underlying data.

**Financial support.** This work was supported by the Wellcome Trust (grant number 100142/Z/12/Z) Fellowship in International Health awarded to L.A. The RISE project is part of the PRogramme for Improving Mental health care (PRIME), which was funded by the UK Department for International Development for the benefit of LMIC (HRPC10). C.H. receives support from the National Institute of Health Research through a RIGHT grant (NIHR200842) and through an NIHR global health research group on homelessness and severe mental illness in Africa (HOPE; NIHR134325). The views expressed in this publication are those of the authors and not necessarily those of the NHS, the National Institute for Health and Care Research or the Department of Health and Social Care, England. C.H. is also funded by the Wellcome Trust through grants 222154/Z20/Z (SCOPE) and 223615/Z/21/Z (PROMISE). H.A.W. receives support from the UK Medical Research Council (MRC) and the UK Foreign Commonwealth and Development Office (FCDO) under the MRC/FCDO Concordat agreement which is also part of the EDCTP2 programme supported by the European Union (MR/R010161/1). The funder of the study had no role in study design, data collection, data analysis, data interpretation or writing of the report.

**Competing interest.** The authors declare that they have no competing interests.

**Ethics standard.** Ethics approvals were obtained from the London School of Hygiene and Tropical Medicine Research Ethics Committee (0735-2), Addis Ababa University College of Health Sciences Institutional Review Board (083/13/Psy) and the Ethiopian National Research Ethics Review Committee (310/048/2015). Written informed consent was sought from each eligible

participant and caregiver by a trial nurse. If a person with schizophrenia was deemed by the nurse not to have capacity to consent, permission was sought from the caregiver and assent from the person with schizophrenia. Individuals were not recruited if they expressed unwillingness to participate. Where the participant was unable to write, a thumb impression was recorded, along with a witness' signature.

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
