## [Reviewer Report]

Dear Professors Dixon Chibanda and Judy Bass,

Re: Submission of manuscript “Community-based Rehabilitation Intervention for people with Schizophrenia in Ethiopia (RISE) cluster-randomized controlled trial: an exploratory analysis of impact on food insecurity, underweight, alcohol use disorder and depressive symptoms ”

Thank you for considering our research article for publication in Global Mental Health. Psychosocial interventions are critical to address the complex social and clinical needs and to achieve functional recovery in persons with schizophrenia. Community-based rehabilitation (CBR) is recommended by the World Health Organisation for resource-poor settings but lack of rigorous evidence from low-income countries is a barrier to policy uptake and scale-up. 

This manuscript presents results of an exploratory analysis from the RISE (Rehabilitation Intervention for people with Schizophrenia in Ethiopia): a Wellcome Trust-funded 12-month cluster-randomised controlled trial conducted in rural Ethiopia, involving 166 participants in 48 sub-districts (NCT02160249). 

To our knowledge, the RISE trial was the first to evaluate the effectiveness of any psychosocial intervention for persons with schizophrenia in a low-income country. We have previously published the main analysis (Asher et al, Lancet Global Health 2022, 10(4):e530-e542). We found that CBR delivered by lay workers, combined with primary care-based mental health care, was effective in reducing disability, symptom severity, caregiver tension and worrying, and in increasing attendance to facility-based care for mental health amongst people with schizophrenia. 

In the current paper, we found that CBR did not have any impact on food insecurity, underweight, alcohol use disorder or depressive symptoms. Despite the negative findings, this study represents a useful contribution to the field of global health. Our findings will help to guide the development and implementation of similar models of care in other low-income countries. We hope you will consider this research article for publication.

Yours sincerely, 

Dr Laura Asher

---

## [Reviewer Report]

The authors have discussed the impact of a cluster RCT of CBR delivered by lay health workers + facility-based care Vs. facility-based care alone for persons with schizophrenia on food insecurity, underweight, alcohol use disorder, and depressive symptoms. This is in addition to the team’s earlier manuscript (Asher et al., 2022) which reported better access to free psychotropic medications, better medication adherence, and lower disability at 12 months in CBR + facility-based care arm Vs facility-based care arm alone. Lay health workers delivered the following interventions: home visits, appointment reminders, and community resource mobilization (mainly for transport to facility-based care + free psychotropic medications). The nature of resource mobilization (in kind or in money), who funded it (public or leaders who may have more resources), and from where (from their own pocket or government funds) is not specified. Most lay health workers did not facilitate public talk by persons with schizophrenia, or conduct family support meetings, or identify or facilitate employment opportunities. It is not clear if the lay health workers mobilized community resources for food and social support for families of persons with schizophrenia. The nature of public awareness-raising meetings, their target audience, and the impact of such meetings are not clear.

The authors may clarify the following aspects for the reader’s benefit.

1. From this and previous manuscripts published by the team, it appears that the lay health worker largely delivered interventions to facilitate continued medical care by psychoeducation, appointment reminders, help in transport, and access to free psychotropic medications (health component of the WHO CBR matrix). This may actually explain why the study didn’t find any impact on food insecurity, underweight, alcohol use disorder, and depressive symptoms. The authors need to specify interventions offered to address education, livelihood, social, and empowerment components of the WHO CBR matrix. The challenges faced in addressing these domains (especially livelihood) need to be discussed.

2. In the study, 11 lay health workers were assigned to deliver CBR to 79 persons with schizophrenia. The work profile of the lay health worker (full-time vs. part-time job for implementing CBR/ wages paid for implementing CBR compared to their earlier jobs) can influence their motivation which will impact the effectiveness of the intervention delivered. The authors may share these details.

3. While authors have elaborated on medical and psychological interventions, they may also discuss social interventions (for poverty alleviation, hunger, and livelihood including microfinance through self-help groups) for persons with severe mental illness in lower and middle-income countries and implications for low-income countries.

4. The wages paid to lay health workers in comparison to money mobilized by them from the community for transport/ free medications may be compared.

5. The support offered by the government in rural Ethiopia in terms of food subsidies, and employment opportunities will help the reader understand the social context.

---

## [Reviewer Report]

Thank you for the opportunity to review “Community-based Rehabilitation Intervention for people with Schizophrenia in Ethiopia (RISE) cluster-randomized controlled trial: an exploratory analysis of impact on food insecurity, underweight, alcohol use disorder and depressive symptoms”. The paper is well-written, and focuses on a neglected area of research into the effectiveness of community-based interventions for severe mental illness. It is encouraging to see the publication of a study with non-significant results, a desperately important and gap in global mental health research. Here are some minor comments:

- In the Introduction, the statement in line 9-13 requires substantiation with a citation.

- The inclusion criteria include having a caregiver, though it could be argued that people without a caregiver might have gotten most from a CBR intervention. In addition, as much of international evidence suggests, the roles of families are paramount in alleviating schizophrenia outcomes – was there perhaps space in the analysis for an adjustment to the degree of family involvement?

- The authors suggest that a dosage of 12 months might have been more effective – is this period a guess, or based on the lengths of exposure to CBR in other settings that have show promise (e.g. it has been shown that interventions such as CBT and CTI significantly increase in effectiveness, the longer the dosage.

- Was there an adjustment in terms of exposure to different models? Or variations in socioeconomic levels of the household?

- A key rationale for the focus of the of the paper relates to the prevention of early mortality. While this no doubt is an important consideration in CBR, the causal linkages between comorbid depression, underweight and food insecurity is not are not entirely clear. Further, there is a danger that quality of life and overall wellness becomes neglected when the focus of interventions are too focused on the prevention of mortality. This is just a general comment, and not necessarily a critique on this paper in particular.

- It might be worth explore the underlying causal reasons why the factors outlined might affect people with schizophrenia in Ethiopia. The socioeconomic status of people might help explain the underweight and food insecurity, and the use of alcohol and depressive symptoms have been described before in Ethiopia, for instance:

The prevalence estimate of depression among people with schizophrenia was found to be 18.0% [95% confidence interval: 14.50–22.30]. Our multivariable analysis revealed that current substance use (AOR 2.28, 95%CI (1.27, 4.09), suicide attempt (AOR 5.24, 95%CI (2.56, 10.72), duration of illness between 6 and 10 years (AOR 2.09, 95%CI (1.08, 4.04) and poor quality of life (AOR 3.13, 95%CI (1.79, 5.76) were found to be the factors associated with depression among people with schizophrenia.

Fanta, T., Bekele, D. & Ayano, G. The prevalence and associated factors of depression among patients with schizophrenia in Addis Ababa, Ethiopia, cross-sectional study. BMC Psychiatry 20, 3 (2020). https://doi.org/10.1186/s12888-019-2419-6

Differences in terms of depression among people with schizophrenia – whether its intrinsic to the illness, related to medication regimens, an expression of negative symptoms etc. – might be important to understand in order to better interpret the results.

- The computer programme used to randomise should be named and cited.

- There are technical errors in some citations.

---

## [Reviewer Report]

Dear authors, this is an article of great scientific and global mental health interest, especially considering the serious consequences for the people affected.

The introduction and methodology described is clear and robust, which allows for a complete overview of this complex phenomenon.

In this regard I have the following minor comments, in order to further improve your article:

Introduction:

1. In relation to the theory of change referred to, although the previous references are cited, could you include a figure or diagram that shows it in graphic form? This in order to be able to capture the attention and logical structure of your intervention, as well as the key points/intermediates, pathways and construction of it.

Methodology:

For the Patient Health Questionnaire (PHQ-9) and the Alcohol Use Disorders Identification Test (AUDIT) do you have information on any psychometric properties of this sample or previous similar sample in the country? (For example, reliability assessed through the use of Cronbach’s alpha coefficient).

Discussion:

-Given the lack of results on the variables, it would be important to be able to discuss about the theory of change posed and see where the assumptions or pathways could explain this lack of effect.

-Another limitation of the study was not including variables associated with the professionals implementing the intervention that could explain differences in the results.

-In the same sense, it would be very interesting and relevant considering the science of implementation, that the authors could discuss and propose as a future line of research in this area and programs, to consider the vision of the participants and professionals in charge of implementation, not only from a quantitative methodological approach, but also from a qualitative perspective, which would allow exploring the possible barriers, facilitators and mechanisms associated with the effect/non-effect of the intervention on the various outcomes and variables of interest.

Others:

1. Review the structure of the references in the text, since in several places the period prior to the citation appears in parentheses, for example “. (Reference)” instead of “(Reference).”.